# Transcoronary Gradients of Mechanosensitive MicroRNAs as Predictors of Collateral Development in Chronic Total Occlusion

**DOI:** 10.3390/medicina60040590

**Published:** 2024-04-03

**Authors:** Mustafa Gökhan Vural, Hulya Yilmaz Temel, Ezgi Turunc, Ramazan Akdemir, Ersan Tatli, Mustafa Tarik Agac

**Affiliations:** 1Department of Cardiology, University of Health Sciences, Ankara Training and Research Hospital, Ankara 06230, Turkey; 2Department of Bioengineering, Faculty of Engineering, Ege University, Bornova 35040, Turkey; hulya.yilmaz.temel@ege.edu.tr; 3Department of Biochemistry, Faculty of Pharmacy, Izmir Katip Celebi University, Izmir 35620, Turkey; ezgi.turunc@ikcu.edu.tr; 4Department of Cardiology, Faculty of Medicine, Sakarya University, Sakarya 54050, Turkey; ramazanakdemir@gmail.com (R.A.); tarikagac@gmail.com (M.T.A.)

**Keywords:** chronic total occlusion, collateral flow index, mechanosensitive microRNAs, transcoronary gradient, coronary collateral development, cytokines

## Abstract

*Background and Objectives*: In this present study, we investigated the impact of mechanosensitive microRNAs (mechano-miRs) on the collateral development in 126 chronic total occlusion (CTO) patients, selected from 810 undergoing angiography. *Materials and Methods*: We quantified the collateral blood supply using the collateral flow index (CFI) and assessed the transcoronary mechano-miR gradients. *Results:* The patients with favorable collaterals had higher CFI values (0.45 ± 0.02) than those with poor collaterals (0.38 ± 0.03, *p* < 0.001). Significant differences in transcoronary gradients were found for miR-10a, miR-19a, miR-21, miR-23b, miR-26a, miR-92a, miR-126, miR-130a, miR-663, and let7d (*p* < 0.05). miR-26a and miR-21 showed strong positive correlations with the CFI (r = 0.715 and r = 0.663, respectively), while let7d and miR-663 were negatively correlated (r = −0.684 and r = −0.604, respectively). The correlations between cytokine gradients and mechano-miR gradients were also significant, including Transforming Growth Factor Beta with miR-126 (r = 0.673, *p* < 0.001) and Vascular Endothelial Growth Factor with miR-10a (r = 0.602, *p* = 0.002). A regression analysis highlighted the hemoglobin level, smoking, beta-blocker use, miR-26a, and miR-663 as significant CFI determinants, indicating their roles in modulating the collateral vessel development. *Conclusions*: These findings suggest mechanosensitive microRNAs as predictive biomarkers for collateral circulation, offering new therapeutic perspectives for CTO patients.

## 1. Introduction

Chronic total occlusion (CTO) stands as a formidable clinical entity within the spectrum of coronary artery disease (CAD), characterized by the enduring occlusion of a coronary artery for a duration extending beyond three months [1]. The pathological significance of CTOs is underpinned by its potential to precipitate ischemia, diminishing quality of life and presenting as a conundrum in cardiovascular therapeutics due to the challenges it poses for medical and surgical management [2]. In response to the ischemic insult of CTOs, the heart initiates the formation of collateral vessels—a natural bypass system crucial for preserving myocardial perfusion [3]. These vessels are a testament to the body’s remarkable capacity for self-preservation in the face of vascular compromise. The genesis and functionality of these collateral networks are modulated by an intricate molecular ballet involving various microRNAs (miRs) and cytokines, which are central to the orchestration of cardiovascular pathology [4,5]. MicroRNAs, small noncoding genes, play a crucial role in post-transcriptionally regulating gene expression by targeting messenger RNA [6].

Atherosclerosis often develops in arterial areas experiencing a disturbed blood flow, leading to gene expression changes [7]. Research shows that changes in flow conditions can affect miR expression in endothelial cells. These flow-sensitive miRs, termed mechano-sensitive microRNAs (mechano-miRs), are key in regulating endothelial gene expression and are linked to endothelial dysfunction and atherosclerosis [8]. They influence critical pathways in endothelial cells, including the cell cycle, inflammation, apoptosis, and nitric oxide signaling [8,9].

In this investigation, mechano-miRs were the focal point due to their unique regulatory role in the response of endothelial cells and cardiovascular pathologies to mechanical stress. Endothelial cells, particularly within coronary arteries, are continually subjected to dynamic alterations in the blood flow [6,7,8,9,10]. Mechano-miRs can modulate the endothelial function and gene expression in response to these mechanical changes, thus playing critical roles in the pathophysiological processes associated with the initiation and progression of atherosclerosis. In conditions such as CTOs, these changes in the blood flow become more pronounced, and mechano-miRs have the potential to influence cardiovascular responses under such stressed states. These microRNAs can affect cellular pathways that are known to be important for processes like the collateral vessel formation and function, as these processes are tightly linked to the adaptive responses displayed by endothelial cells under mechanical stress. Therefore, examining the role of these mechano-miRs in understanding the collateral vessel development and in identifying potential therapeutic targets in CTOs represents a significant step in the management of cardiovascular diseases.

A critical aspect of understanding the pathophysiology of a CTO and its impact on the collateral development is the study of the transcoronary gradients of mechano-miRs [11,12]. The transcoronary approach enables the precise delineation of the endogenous factor that directly contributes to the formation of collateral vessels and the maintenance of myocardial perfusion [5,11,12]. It is particularly pertinent in the context of a CTO, where the differential between the levels of mechano-miRs in the coronary arterial and venous blood reflects the active biological processes across the occlusion site. The transcoronary gradients of mechano-miRs serve as a beacon, illuminating the path of the collateral vessel development amid the turbulent waters of a CTO. They offer a window into the delicate balance of pro- and anti-inflammatory forces, angiogenesis, and tissue remodeling that converge to dictate the fate of the ischemic myocardium. While previous research has cast light upon the roles of mechano-miRs in ischemic heart disease, a comprehensive understanding of their transcoronary activities in CTOs remains elusive.

This study has the potential to make significant contributions to the field of cardiovascular medicine by providing a deeper understanding of the biological underpinnings of the collateral vessel development in CTOs. By exploring the central role of transcoronary gradients, we aim to unveil novel biomarkers and therapeutic targets that could revolutionize the management strategies for CTOs, ultimately enhancing patient outcomes and advancing the frontier of cardiovascular healthcare.

## 2. Materials and Methods

### 2.1. Study Design

In this two-center observational study, the association between mechano-miRs and the collateral vessel development in CTO patients was examined. The objective of this study was to understand the molecular mechanisms driving collateral circulation in CTOs. Of 810 patients undergoing coronary angiography, 126 were selected based on stringent criteria. These included excluding those with a history of myocardial infarction, a left ventricular ejection fraction below 50%, or a previous coronary artery bypass grafting. Additionally, exclusion criteria encompassed an age over 75, uncontrolled diabetes mellitus, uncontrolled hypertension, chronic inflammatory states, active malignancy, significant liver or kidney dysfunction, recent acute coronary syndrome, and notable hematological imbalances (Figure 1). This careful selection ensured a homogenous group for accurate cardiac analysis. The study population was categorized into three groups: 30 subjects without significant coronary artery disease (Group I), 33 with CAD but no CTO (Group II), and 63 with a CTO (Group III).

### 2.2. Catheterization and Transcoronary Gradient Assessment

Advanced catheterization techniques were employed to collect blood samples from the aortic root and the right atrium or coronary sinus, enabling the measurement of the transcoronary gradients [4,5,11,12,13]. These gradients represent the difference in mechano-miR and various cytokine concentrations between the arterial and venous samples, offering insights into the molecular interplay across the coronary circulation.

The collateral flow index (CFI) is a method used to quantify the effectiveness of the collateral blood flow in the heart [5]. The method involves measuring pressures in specific coronary regions, both in normal and occluded states with a dedicated intracoronry guidewire (PressureWire™ X Guidewire, St Jude Medical, Chicago, IL, USA). The index is calculated using the following formula: *CFI* = (*Pa* − *Pv*)/(*Poccl* − *Pv*) [12,13]. In this formula, *Poccl* represents the coronary occlusive pressure, *Pa* is the aortic pressure, and *Pv* is the central venous pressure. The aortic and venous pressures are measured, and the coronary occlusive pressure is obtained by temporarily occluding the coronary artery during the coronary angiography. The index provides a quantitative assessment of the coronary collateral circulation’s ability to supply blood to the myocardium during a coronary artery occlusion. A higher CFI indicates better collateral circulation [12,13]. In our investigation, ethical considerations precluded the execution of transcoronary gradient measurements within the cohort of healthy volunteers.

The Rentrop classification system was utilized to categorize the collateral vessel development based on the angiographic findings [14]. Myocardial ischemia was evaluated using a myocardial perfusion scintigraphy. This method allowed for the classification of ischemia severity into mild, moderate, and severe categories, providing a nuanced understanding of the myocardial condition in the context of CTOs [15].

### 2.3. Mechano-miR and Cytokine Analysis

The blood samples were collected from patients during a cardiac catheterization. The proper handling and immediate refrigeration of the samples were ensured to maintain the integrity of the mecanho-miRs and various cytokines. The cytokine levels were quantified using the sandwich enzyme-linked immunosorbent assay [16]. The total RNA, including microRNAs, was extracted from blood samples using a commercially available kit (PAXgene Blood microRNA Kit, Cat. No: 763134, Qiagen, Hilden, Germany). This kit is specifically designed for high-quality RNA extraction, crucial for an accurate microRNA analysis [17]. The extracted RNA was reverse-transcribed to cDNA using specific primers for the target mechano-miRs. A quantitative real-time PCR was conducted using TaqMan assays. This step amplified specific mechano-miRs and allowed for their quantification. A panel of mechano-miRs known to play roles in vascular remodeling, angiogenesis, inflammation, and endothelial function was profiled. The relative expression levels of the mechano-miRs were calculated using the 2^−ΔΔCt^ method [18]. The snRNA (SNORD61) included in the kit was used as the reference microRNA.

### 2.4. Statistical Analysis

This study employed a combination of parametric and non-parametric tests for the data analysis, ensuring the appropriateness of the statistical methods to the data distribution. In addition to descriptive statistics, correlation analyses were conducted to explore the relationships between transcoronary gradients of mechano-miRs and the extent of the collateral vessel development. Multiple stepwise linear regression was used to identify predictors of the CFI, considering demographic and clinical variables as potential confounders. A power analysis, adhering to clinical research norms with effect sizes and an alpha level of 0.05, underscored the need for at least 18 participants in each group to robustly ascertain the impact of mechano-miRs on the collateral development in CTO patients. Consequently, a meticulously designed approach led to the evaluation of 120 CTO patients, aligning with our catheterization labs’ capabilities, and facilitated the review of 810 consecutive angiographies, achieving an 80% power threshold for conclusive insights. The statistical significance was set at a *p*-value of <0.05. This study was conducted in accordance with the highest ethical standards, adhering to the principles outlined in the Declaration of Helsinki. Approval was obtained from the relevant Institutional Review Boards, and informed consent was secured from all participants.

## 3. Results

### 3.1. Demographic and Clinical Characteristics of Study Cohort

A meticulous examination of the demographic, clinical, and biochemical parameters across the study groups has unveiled numerous pivotal observations, as comprehensively detailed in Table 1. Firstly, the baseline demographic factors including age, gender, body mass index, hypertension, dyslipidemia, diabetes mellitus prevalence, and smoking status exhibited no significant differences across the groups. Similarly, the biochemical parameters such as hemoglobin, fasting glucose, lipid profiles, creatinine, and leukocyte counts were uniformly distributed, indicating no significant disparities. However, notable differences were observed in several clinical and biochemical markers. Aspirin usage was significantly more prevalent in the CAD and CTO groups compared to the healthy subjects, reflecting its routine clinical application in these conditions (*p* < 0.001). A significant gradation in beta blocker usage was evident, increasing from healthy individuals to patients with CTOs (*p* = 0.006). The prevalence of diastolic dysfunction was higher in the CTO group (*p* = 0.014), suggesting a potential link with chronic occlusive coronary pathology. A symptom duration exceeding three months was considerably more common in the CTO patients, indicative of the chronic nature of their condition (*p* < 0.001). An ischemic burden quantified as less than 5% was predominantly seen in the healthy subjects (*p* < 0.001).

Biochemically, a significant upregulation of high-sensitive Troponin (hs-Troponin) and Matrix Metalloproteinase-2 (MMP-2) levels was observed in the CTO patients (*p* = 0.032 and *p* = 0.016, respectively), underscoring the ongoing myocardial stress and a sustained ischemic injury response. Conversely, the Vascular Endothelial Growth Factor (VEGF) levels were notably lower in the CAD patients, including those with a CTO (*p* = 0.001), potentially reflecting a disrupted endothelial response in chronic coronary syndromes. The pro-inflammatory cytokine Interleukin-1β (IL-1β) was significantly elevated in the CTO patients (*p* = 0.042), indicating its role in the inflammatory cascade characteristic of CTO pathology.

### 3.2. Transcoronary MicroRNA and Cytokine Profiles: Their Association with the Collateral Flow Index in Chronic Total Occlusion Patients

A comparative analysis among the CTO patients distinguished by favorable versus poor collateral circulation is detailed in Table 2. The CFI was significantly higher in the favorable collateral group compared to the poor collateral group (0.45 ± 0.02 vs. 0.38 ± 0.03; *p* < 0.001). Age, gender, hypertension, diabetes mellitus, dyslipidemia, smoking status, and the use of RAAS inhibitors, statins, beta blockers, calcium channel blockers, and acetylsalicylic acid did not show significant differences between the groups with favorable and poor collateral circulation. This suggests that these factors are not distinctly associated with the quality of collateral circulation in patients. Additionally, no significant difference was observed in diastolic dysfunction and the location of culprit lesions between the groups.

The left ventricular ejection fraction, hemoglobin levels, leukocyte and platelet counts, neutrophil and lymphocyte counts, and the neutrophil-to-lymphocyte ratio did not exhibit significant differences between the groups. However, higher glucose levels were noted in the group with poor collateral circulation, suggesting a potential link with the metabolism of glucose. The creatinine levels and markers of cardiac injury and inflammation, including hs-Troponin and hs-CRP, along with the lipid profiles, showed no significant association with the quality of collateral circulation.

Notably, significant differences were observed in the transcoronary gradients of mechano-miRs and various cytokines, with TGF-β and miR-126 showing a substantial disparity between the groups (*p* = 0.003 and *p* < 0.001, respectively). Moreover, the gradients of miR-10a, miR-19a, miR-21, miR-23b, miR-26a, miR-92a, miR-130a, miR-663, and let7d were all significantly associated with the collateral flow quality (*p* < 0.05 for all), suggesting that these biomarkers may play crucial roles in the development or function of collateral circulation. VEGF showed a significant transcoronary gradient difference between the two groups (*p* = 0.006).

### 3.3. Multivariate Analysis of Factors Influencing the Collateral Flow Index in Chronic Total Occlusion Patients

The correlations between the transcoronary gradients of mecahno-miRs and cytokines with the CFI are comprehensively elucidated (Figure 2, Figure 3, Figure 4, Figure 5, Figure 6, Figure 7 and Figure 8). The analysis revealed that increases in the transcoronary gradients of the TGF-β and VEGF were moderately negatively correlated with the CFI (r = −0.419 and r = −0.397, respectively, with *p*-values of 0.021 and 0.030). Among the mechano-miRs studied, ∆ miR-10a exhibited a moderate negative correlation with the CFI (r = −0.462, *p* = 0.023), which could implicate this miR in the inhibition of collateral circulation. Conversely, ∆ miR-19a displayed a moderate positive correlation (r = 0.471, *p* = 0.020), aligning with a beneficial influence on the development of collateral vessels. Notably, ∆ miR-21 and ∆ miR-26a showed strong positive correlations with the CFI (r = 0.663, *p* = 0.001 and r = 0.715, *p* < 0.001, respectively), indicating a significant association with an enhanced collateral flow. These findings suggest that higher gradients of miR-21 and miR-26a across the coronary circulation might play a pivotal role in promoting the growth or function of the collateral vessels. On the other hand, ∆ miR-126 and ∆ miR-663 were identified to have moderate negative correlations with the CFI (r = −0.481, *p* = 0.017 and r = −0.604, *p* = 0.002, respectively), hinting at a possible inhibitory role in collateral circulation. The strongest negative correlation was observed with ∆ let-7d (r = −0.684, *p* < 0.001), indicating a substantial inverse relationship with the CFI.

In our study, a sophisticated interplay was uncovered between the transcoronary gradients of mechano-miRs and cytokines, highlighting their concerted impact on the development of collateral vessels (Figure 9, Figure 10, Figure 11, Figure 12, Figure 13 and Figure 14). The venous levels of hs-CRP, a marker of systemic inflammation, show significant positive correlations with the transcoronary gradients of miR-21 (r = 0.500, *p* = 0.013) and miR-26a (r = 0.425, *p* = 0.038). These associations point to a potential link between the systemic inflammatory status and the modulation of miR expression across the coronary circulation, which in turn may influence the development and function of the collateral vessels. The negative correlations of ∆ TGF-β with ∆ miR-21 (r = −0.418, *p* = 0.042) and ∆ miR-26a (r = −0.436, *p* = 0.033) suggest that the regulatory effects of TGF-β on collateral flow are, at least in part, mediated through changes in the levels of these miRs. The negative correlation with miR-21 and miR-26a could indicate that higher TGF-β levels might inhibit the expression of these miRs, which have been shown to be positive regulators of collateral flow. The robust positive correlation between ∆ TGF-β and ∆ miR-126 (r = 0.673, *p* < 0.001) stands out, reinforcing the notion that TGF-β might have a complex, context-dependent role in modulating collateral circulation. VEGF, a critical pro-angiogenic factor, exhibits strong correlations with the mechano-miR gradients, most notably with ∆ miR-10a (r = 0.602, *p* = 0.002) and inversely with ∆ miR-21 (r = −0.518, *p* = 0.010). The positive correlation with ∆ miR-26a (r = 0.516, *p* = 0.010) further confirms the pro-angiogenic role of this mechano-miR, potentially mediated through the VEGF signaling pathways. Furthermore, IL-1β, a pro-inflammatory cytokine, shows a positive correlation with ∆ miR-92a (r = 0.436, *p* = 0.033) and a marginally significant correlation with ∆ miR-663 (r = 0.402, *p* = 0.052). This association could reflect the complex interplay between inflammation and the miR-mediated regulation of collateral circulation. Collectively, these correlations validate the profound impact of miRs on collateral flow, as modulated by cytokines associated with angiogenesis and inflammation.

In our study, a detailed multivariate analysis was conducted to elucidate the relationship between various biological factors and their impact on the CFI. The analysis incorporated several models, each progressively including different variables to assess their individual and combined effects on the CFI. The transcoronary gradient of miR-26a showed a notable positive impact on the CFI (B = 0.001, SE = 0.000, Beta = 0.541, t = 4.914, *p* = 0.000, 95% CI [0.000, 0.001]), indicating its crucial role in promoting collateral circulation. In contrast, ∆ miR-663 was inversely correlated with the CFI (B = −0.001, SE = 0.000, Beta = −0.546, t = −4.717, *p* = 0.000, 95% CI [−0.001, 0.000]), suggesting a potential inhibitory effect on the collateral vessel formation. Additionally, the use of beta blockers appeared to negatively influence the CFI (B = −0.029, SE = 0.009, Beta = −0.359, t = −3.240, *p* = 0.004, 95% CI [−0.047, −0.010]), while higher levels of hemoglobin were associated with an improved CFI (B = 0.004, SE = 0.002, Beta = 0.234, t = 2.111, *p* = 0.048, 95% CI [0.000, 0.008]). These findings, encapsulated in the statistical significance and confidence intervals, provide a comprehensive understanding of the factors influencing the development of collateral vessels in chronic total occlusion, highlighting the complex interplay of molecular and clinical factors.

## 4. Discussion

In this study, we explored the contributory role of mechano-miRs in the formation of collateral vessels within the context of a CTO. Our analysis delved into the transcoronary gradients of mechano-miRs and assessed their impact on the collateralization of the vascular network in patients with a CTO, while also considering the intermediary influence of various cytokines. These mechano-miRs were specifically selected due to their pivotal role in modulating the endothelial responses to mechanical stress—an aspect fundamental to the pathophysiological understanding of CTOs. The insights gained from this research have the potential to significantly refine CTO management strategies by pinpointing novel mechano-miR and cytokine targets for therapeutic innovation, ultimately aiming to improve clinical outcomes for patients.

Our study unveiled substantial correlations between the transcoronary gradients of certain mechano-miRs, specifically miR-21 and miR-26a and the CFI. These correlations suggest that the elevated gradients of these mechno-miRs across the coronary system are linked to an enhanced collateral flow. Conversely, negative correlations with the miR-663 and let-7d gradients imply a possible inhibitory impact on the development or functionality of collateral vessels. Additionally, our research investigated the interplay between transcoronary cytokine gradients and the development of collateral vessels. VEGF’s role in promoting angiogenesis is well-documented, while TGF-β’s involvement in vascular remodeling and angiogenesis is recognized as more complex. The identified significant correlations between these cytokine gradients and mechano-miRs, particularly miR-26a, highlight the intricate and concerted molecular interactions that facilitate the formation of collateral vessels, suggesting a cooperative mechanism of action in angiogenic processes. These findings are particularly relevant in the context of cardiovascular therapeutics, where the enhancement of the formation of collateral vessels can mitigate the ischemic consequences of CTOs and improve patient outcomes [19]. The identification of specific mechano-miRs and cytokines as potential biomarkers for the collateral vessel development opens new avenues for the development of targeted therapies aimed at modulating these molecular pathways.

The demographic and clinical profiles within our study cohort provide a broad perspective on the foundational characteristics of these distinct patient groups. The lack of significant differences in age and gender among the groups supports the existing literature, suggesting that these basic demographic elements do not singularly predict CTO occurrence or CAD severity [20]. This aligns with the understanding that CAD and CTOs are influenced by a spectrum of factors beyond straightforward demographic classifications. Interestingly, the prevalence of hypertension and dyslipidemia, well-documented risk factors for CAD, did not present statistically significant differences across the groups [21]. This may imply that, while these conditions contribute to the initial onset of CAD, their impact on progressing to a CTO might be tempered by other specific pathophysiological processes, such as inflammatory responses or endothelial health. The uniform distribution of diabetes and the nonsignificant differences in smoking habits further underscore the intricate interplay of risk factors in CAD and CTO progression. These patterns highlight the necessity for a comprehensive approach to risk assessment and treatment in these conditions, which extends past conventional risk factor models. The widespread use of aspirin and beta blockers in the CAD and CTO patient groups mirrors established clinical protocols for CAD, reinforcing their importance in secondary prevention. Furthermore, the association of diastolic dysfunction and prolonged symptom duration with the CTO patients reflects the persistent nature of the condition and its effect on cardiac function. Elevations in hs-Troponin and MMP-2 within the CTO patients point to persistent myocardial stress and structural heart remodeling, shedding light on the active pathobiology of CTOs and indicating a continuous myocardial response to long-term ischemic injury [22,23]. The lower levels of VEGF observed in the CAD and CTO patients might indicate a compromised angiogenic response, which is essential for the collateral vessel development and represents a potential focus for therapeutic intervention in CTO treatment [24,25,26]. Our comparative analysis between the patients with favorable and poor collateral circulation reveals that the formation and quality of the collateral flow in a CTO are not directly governed by traditional cardiovascular risk factors or common pharmacological treatments, marking a significant observation that may guide future clinical strategies.

The comprehensive multivariate regression analysis underscores the predictive significance of specific mechano-miRs and clinical factors in determining the CFI in patients with a CTO. This analysis elucidates the complex influence of therapeutic interventions, systemic physiological parameters, and lifestyle factors on collateral circulation. Notably, this study reveals a nuanced relationship between beta blocker usage and a reduced collateral vessel development, contrasting with some literature suggesting their neutral or even beneficial effects, potentially due to induced peripheral resistance and angiogenic stimulation [27]. This inverse association may reflect the intricate consequences of beta blocker therapy on cardiovascular dynamics, possibly dampening exercise-induced angiogenic responses or affecting hemodynamic forces essential for collateral vessel maturation. Such findings advocate for personalized therapy in CTO management and signal a need to carefully consider the presumed benefits of beta blockers on collateral vessel growth. Furthermore, the analysis confirms the importance of hemoglobin levels in collateral flow, indicating the vital role of oxygen transport in supporting ischemic tissue perfusion [28]. This suggests that therapeutic efforts to manage anemia and optimize oxygen delivery are paramount in improving collateral circulation and patient outcomes in a CTO. Lastly, this study reaffirms the detrimental impact of smoking on the CFI, echoing the widespread recognition of smoking’s negative vascular effects [29]. This emphasizes smoking cessation as an essential strategy for enhancing the collateral vessel development and overall cardiovascular health in CTO patients.

These findings also highlight the importance of considering other potential determinants of collateral circulation quality, notably the role of microRNA gradients across the coronary circulation. MicroRNAs, with their pivotal role in gene regulation, have emerged as crucial mediators of vascular biology, influencing processes such as endothelial function, angiogenesis, and vascular remodeling [8]. The potential for transcoronary microRNA gradients to serve as biomarkers or therapeutic targets for enhancing the collateral vessel development offers an exciting avenue for research and clinical intervention. This is particularly relevant given the lack of significant associations between collateral flow and traditional risk factors or medication usage observed in our study.

In the realm of the coronary collateral development, miR-21 and miR-26a have been identified as significant pro-angiogenic mediators [24,30]. Their elevated levels correlate with enhanced collateral circulation, indicating their crucial roles in modulating angiogenic signaling pathways. These microRNAs do not merely stimulate endothelial cell function; they orchestrate a wider regulatory network that underpins angiogenesis. Their upregulation during ischemic conditions posits a therapeutic opportunity: enhancing their expression could feasibly bolster the collateral vessel formation, thus presenting a novel therapeutic angle in CTO management. However, it is essential to carefully navigate the context-specific effects and potential risks. The overexpression of miR-21, for instance, has been implicated in adverse outcomes such as oncogenesis and fibrotic processes, necessitating targeted and controlled modulation to harness therapeutic benefits while minimizing the possibility of deleterious side effects [31,32]. miR-126 is recognized for its pivotal role in preserving endothelial integrity and facilitating angiogenesis [33,34]. The negative correlation with the CFI in our study introduces a nuanced view into its function, indicating that its regulatory balance is crucial for the collateral vessel development. An excess or deficiency of miR-126 could impede the vascular remodeling necessary for adequate collateralization. Our findings suggest that precise modulation of miR-126 is essential to harness its angiogenic potential without destabilizing the vascular structures. The typical pro-angiogenic role of miR-126 is complicated by our study’s evidence of its possible inhibitory effects under certain ischemic conditions, challenging the anticipated positive correlation with the collateral vessel development. This unexpected relationship signals a sophisticated regulatory landscape in ischemic regions, where an overabundance of miR-126 may hinder effective angiogenesis or the maturation of collateral vessels. The observed discrepancy likely arises from the diverse targets and actions of miR-126, which are potentially modulated by surrounding cytokine profiles, underscoring the significance of the local ischemic context in dictating its overall impact on collateral blood flow. miR-10a is recognized for its regulatory influence on endothelial gene expression and its capacity to attenuate pro-inflammatory responses within the vascular system [35]. Its significant relationship with the collateral flow quality in our study suggests it plays a critical role in mitigating inflammation-induced vascular impairment, thus promoting a more efficient endothelial function and potentially enhancing the collateral vessel development. miR-19a, identified as a component of the miR-17-92 cluster, is noted for its engagement in angiogenic and vascular remodeling activities, influencing mechanisms that foster endothelial cell resilience and proliferation [36]. The positive correlation between miR-19a and the CFI emphasizes its integral role in adaptive vascular responses, vital for counteracting ischemic challenges. miR-23b, miR-92a, and miR-130a’s participation in angiogenesis and vascular remodeling elucidates the intricate roles microRNAs play in the collateral vessel formation. miR-23b’s involvement in endothelial cell proliferation and migration, fundamental to neovascularization, is indicative of its potential to augment ischemic angiogenic responses, as reflected in its correlation with the CFI [8]. Conversely, miR-92a is recognized for its angiogenic inhibitory effects in certain contexts, signifying the necessity for its careful modulation to preserve a harmonious balance between angiogenic stimulators and suppressors within ischemic territories [37]. Meanwhile, miR-130a is noted for mitigating anti-angiogenic influences, thereby fostering vascular development by liberating endothelial cells from inhibitory cues [38]. In the realm of vascular biology and the response to ischemia, miR-663 and let-7d present intriguing differences. miR-663 has been implicated in the modulation of inflammatory responses, potentially affecting vascular remodeling by adjusting the equilibrium between pro-inflammatory and anti-inflammatory cytokines within ischemic tissues [39]. Its inverse relationship with the CFI may signify its function in mitigating undue inflammatory damage, thus indirectly facilitating the formation of collateral vessels. On the other hand, the involvement of let-7d in targeting elements of the angiogenic signaling cascade indicates a subtle role, whereby its expression levels could be pivotal in determining the efficacy of the collateral vessel development in response to ischemic conditions [40]. Our research underscores the detrimental effects of miR-663 and let-7d on the CFI, proposing that these microRNAs might act as barriers to the development of collateral vessels. This observation departs from the general body of microRNA research, which has not extensively associated these microRNAs with collateral circulation in CTOs. The inhibitory actions of miR-663 and let-7d on the collateral vessel formation could be ascribed to their governance over inflammatory responses and angiogenic signaling pathways, potentially engendering a milieu inhospitable for the maturation of collateral vessels [41]. Such a divergence underscores the intricate nexus between inflammation, angiogenesis, and microRNA expression, highlighting the necessity for a nuanced regulatory strategy to augment the development of collateral vessels.

The intricate dynamics among these microRNAs, each bearing unique yet interrelated roles within vascular biology, shed light on the multifaceted nature of the collateral vessel development in the context of chronic total occlusions (CTOs). Delving into the precise molecular mechanisms by which these miRs modulate angiogenesis and vascular remodeling presents promising therapeutic prospects. Future investigations should endeavor to dissect these pathways with greater specificity, examining the potential of targeted microRNA modulation as an innovative approach to bolster collateral circulation in patients afflicted with a CTO. This strategy requires a meticulous equilibrium, leveraging the pro-angiogenic capacities of certain miRs, whilst counteracting the adverse effects of others, thereby cultivating a milieu that favors the efficacious formation and functionality of collateral vessels.

The associations between transcoronary cytokine gradients and the subtle modulation by mechanomiRs provide insight into the complex regulatory mechanisms that underlie vascular biology and myocardial adaptation in the context of CTOs. These intricate relationships reveal a multifaceted interplay, where cytokines and miRs not only reflect myocardial metabolic activity but also actively contribute to the pathological and reparative processes within the ischemic myocardium.

VEGF has long been recognized as a quintessential mediator of angiogenesis, orchestrating the proliferation and migration of endothelial cells [24,25,26]. This angiogenic imperative is critically regulated by an ensemble of microRNAs, as evidenced by the correlations with miR-10a, miR-21, miR-26a, and let-7d. Our data revealed that VEGF’s influence is far from unilateral; it is subject to a delicate calibration by these miRs. The negative correlation with the CFI poses a conundrum, challenging the conventionally held belief of VEGF’s unilaterally beneficial role in promoting vascular integrity and growth. This suggests a paradox where VEGF’s signal, in the absence of precise regulation, may precipitate the development of aberrant or immature vessels, thereby subverting its angiogenic intent. The implication here is profound: therapeutic interventions seeking to harness VEGF’s potential must be finely tuned to not simply enhance its activity but to shape the vascular milieu to encourage the maturation of robust, functional collateral networks. In contrast, IL-10 distinguishes itself as an anti-inflammatory sentinel, promoting vascular health and potentially facilitating collateral vessel growth [42]. Its positive correlation with MMP-2 and implications for the CFI point to a role that extends beyond the mere suppression of inflammation. It suggests a restorative influence, where IL-10 may actively participate in remodeling the extracellular matrix, fostering a conducive environment for the collateral vessel development and maturation [43]. The relationship between IL-1β and miRs such as miR-92a and miR-663, with their nuanced impacts on the CFI, further complicates the narrative. These correlations illustrate a multifaceted role for inflammatory mediators, which may pivot from being detrimental to being indispensable in the angiogenic process, depending on the stage and context of the inflammatory response.

TGF-β is a multifaceted cytokine with a dual role in promoting both fibrosis and angiogenesis [44]. Its regulatory impact is context-dependent; where, in one milieu, TGF-β may foster the fibrotic response, in another, it might encourage angiogenic processes. The negative correlations with ∆miR-21 and ∆miR-26a and the positive correlations with ∆miR-126 and ∆let-7d suggest that TGF-β’s influence on the myocardial environment is nuanced and likely modulated by a network of microRNAs. In the myocardial setting, TGF-β’s promotion of fibrosis can be a reparative response following injury, contributing to the structural integrity of the heart [44]. However, excessive fibrosis can lead to stiffening of the cardiac tissue, impairing its function and potentially compromising the collateral vessel development. This is where the balance between fibrotic and angiogenic signaling becomes critical, particularly in the setting of a CTO where the formation of functional collateral vessels is essential for maintaining myocardial perfusion. The positive correlation between TGF-β and miR-126 could reflect a regulatory mechanism where TGF-β may enhance the expression of miR-126 to support endothelial cell function and angiogenesis, which is vital for the collateral vessel development. miR-126 has been known to support vascular integrity and angiogenesis, and its upregulation by TGF-β could represent a compensatory mechanism in response to ischemic conditions [34]. Conversely, the negative correlation with ∆miR-21 and ∆miR-26a might indicate a suppressive role of TGF-β on these miRs, possibly to curb their pro-angiogenic effects or to regulate their involvement in other pathways, such as inflammation or cell survival [45]. The interplay between TGF-β and the mechano-miRs is a testament to the complexity of the signaling networks within the ischemic myocardium. The balance between TGF-β’s fibrotic and angiogenic actions may be a critical determinant of the myocardial response to ischemia and the success of the collateral vessel formation. Thus, understanding the context-specific roles of TGF-β and its regulation by microRNAs could provide valuable insights for developing therapeutic strategies aimed at enhancing the collateral vessel development in CTOs.

These insights delineate a fertile ground for further inquiry. They beckon a more nuanced interrogation of the molecular interplay at hand, perhaps through longitudinal studies that could capture the temporal evolution of these gradients and their impact on the ischemic myocardium. The ultimate goal would be to translate these molecular dialogues into a lexicon of therapeutic strategies, enabling the precise sculpting of vascular responses to optimize myocardial perfusion in the face of occlusive coronary artery disease.

This study has a limited sample size that restricts its statistical power to detect subtle associations and may not represent the broader population with CTOs. Future studies should aim for larger, more diverse cohorts that include a wide range of demographic and genetic backgrounds to enhance generalizability. This study is conducted in two institutions, and the findings may be influenced by local practice patterns and patient populations. Multi-center studies are recommended to validate findings across different populations and healthcare systems. This study’s cross-sectional design precludes the ability to establish causality or observe temporal changes in mechano-miR expression. Longitudinal studies would allow for the observation of changes over time and a better assessment of cause–effect relationships. The selection of appropriate control groups is vital. If the controls are not well matched for cardiovascular risk factors, the results could be confounded. Future studies should ensure that the control subjects are matched on key clinical and demographic variables. The CFI is one method to assess collateral flow, but it may not fully capture the functional adequacy of the collateral vessels. Incorporating other imaging modalities or functional assessments could provide a more comprehensive evaluation. The methodology for measuring cytokines and mechano-miRs must be precise and reproducible. Any variability in the measurement techniques can lead to inconsistent results. Standardizing these methods and using quality controls can improve reliability. Various unmeasured confounders such as medication adherence, dietary factors, physical activity levels, and psychosocial stressors might affect the mechano-miR levels and collateral vessel formation. Future studies should aim to measure and adjust for these confounders. This study’s findings may not be generalizable to other forms of ischemic heart disease beyond CTOs. Comparative studies in different ischemic contexts are needed to delineate the role of mechano-miRs in a broader spectrum of cardiovascular disease. This study may lack direct mechanistic insights into how mechano-miRs influence the collateral vessel formation. Experimental studies, possibly using animal models or in vitro systems, could provide a deeper understanding of the underlying biological mechanisms. This study does not include intervention trials to modulate mechano-miRs or cytokines. To move from association to clinical application, interventional studies are necessary to determine if modifying these molecules can actually improve patient outcomes.

## 5. Conclusions

This study elucidates the contributory roles of mechano-miRs and intermediary cytokines in the collateral vessel formation in CTOs, highlighting the positive regulatory potentials of miR-21 and miR-26a and the inhibitory effects of miR-663 and let-7d. The interplay between these miRs and cytokines such as VEGF and TGF-β suggests complex regulatory mechanisms that could be targeted in novel therapeutic strategies. Further research with larger cohorts is necessary to confirm these findings and fully translate them into clinical practice.

## Figures and Tables

**Figure 1 medicina-60-00590-f001:**
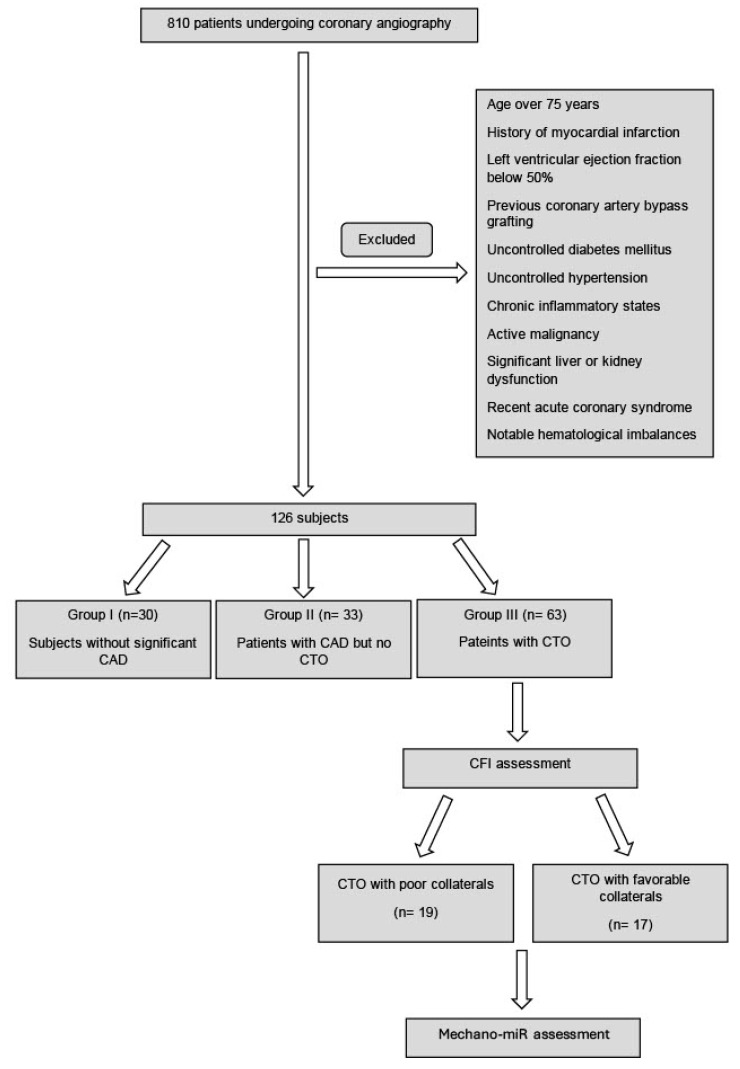
Participant Selection Process Flowchart.

**Figure 2 medicina-60-00590-f002:**
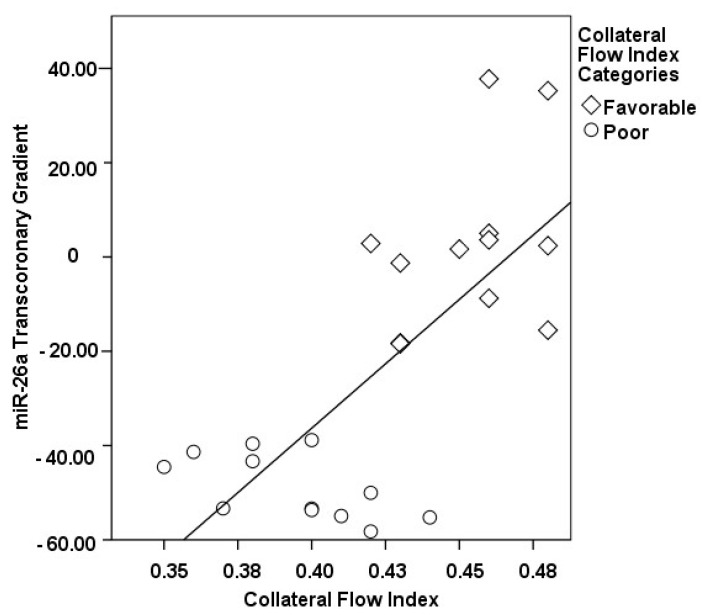
The correlation between the transcoronary gradient of miR-26a and the CFI.

**Figure 3 medicina-60-00590-f003:**
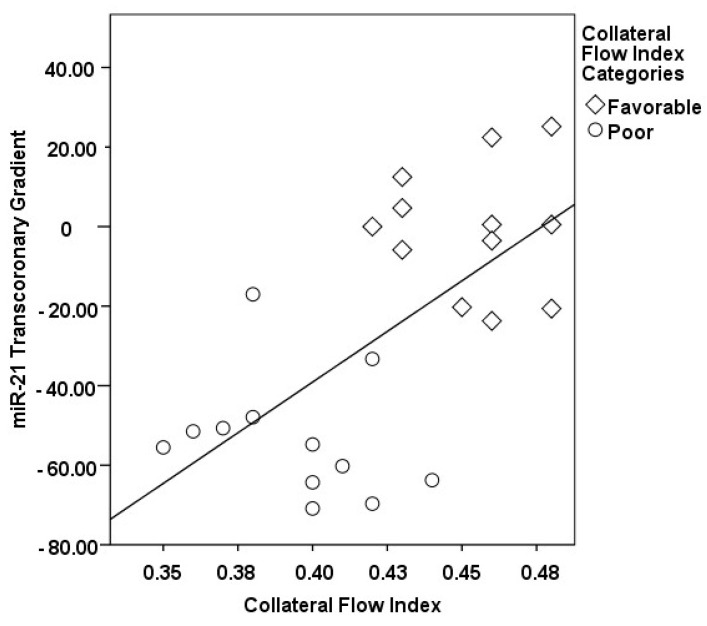
The correlation between the transcoronary gradient of miR-21 and the CFI.

**Figure 4 medicina-60-00590-f004:**
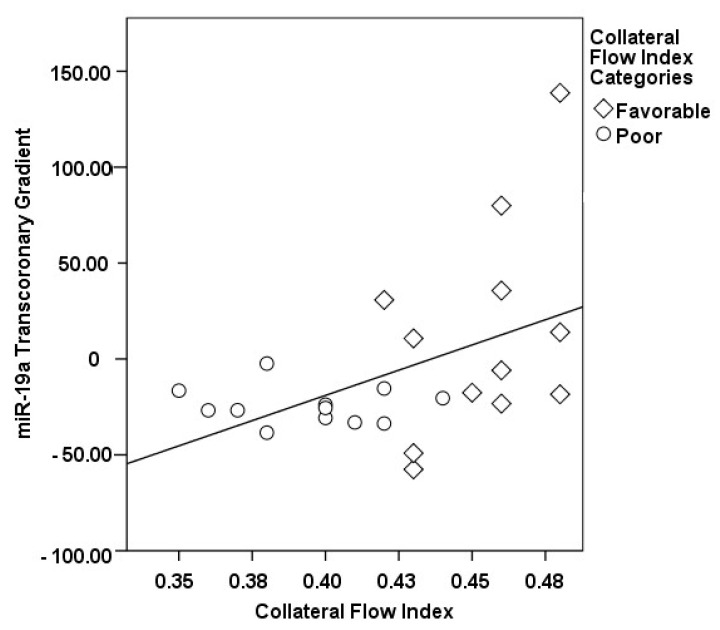
The correlation between the transcoronary gradient of miR-19a and the CFI.

**Figure 5 medicina-60-00590-f005:**
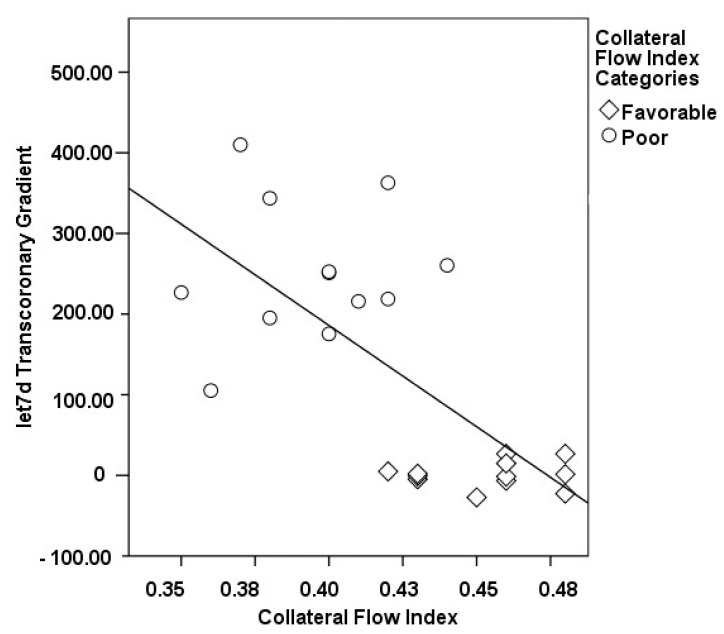
The correlation between the transcoronary gradient of let 7d and the CFI.

**Figure 6 medicina-60-00590-f006:**
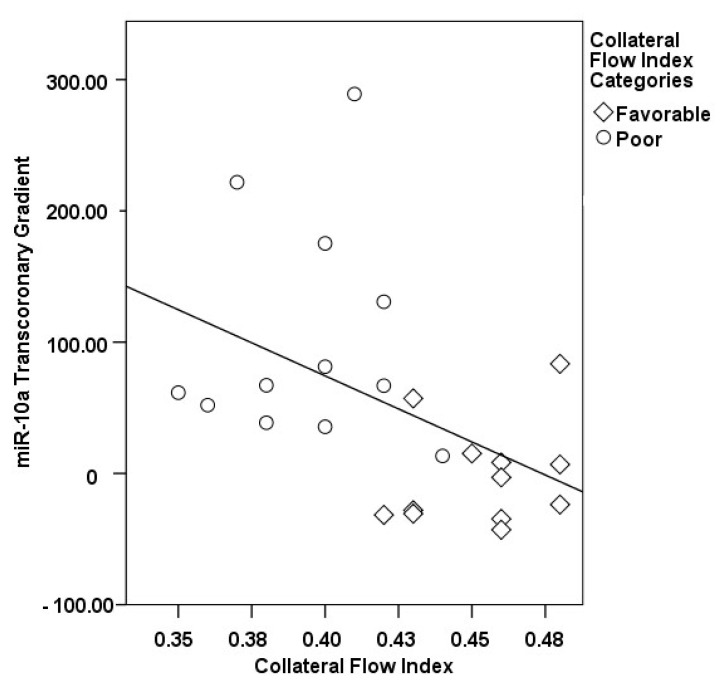
The correlation between the transcoronary gradient of miR-10a and the CFI.

**Figure 7 medicina-60-00590-f007:**
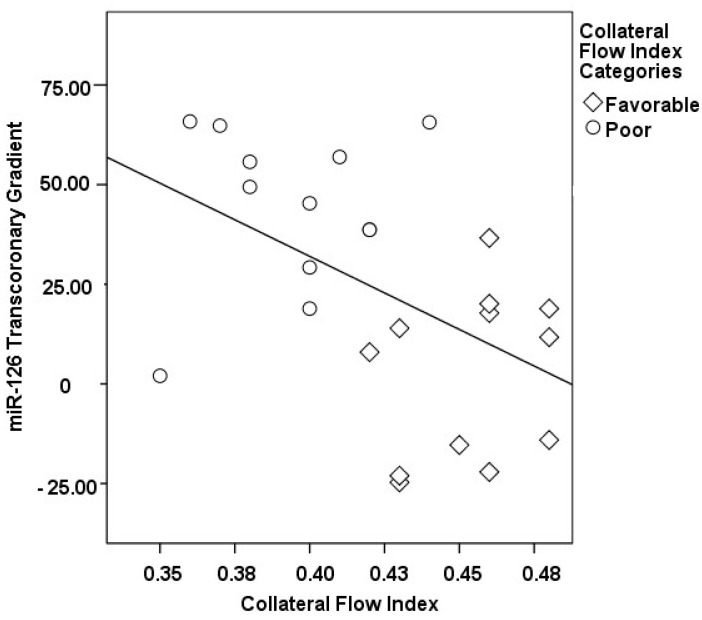
The correlation between the transcoronary gradient of miR-126 and the CFI.

**Figure 8 medicina-60-00590-f008:**
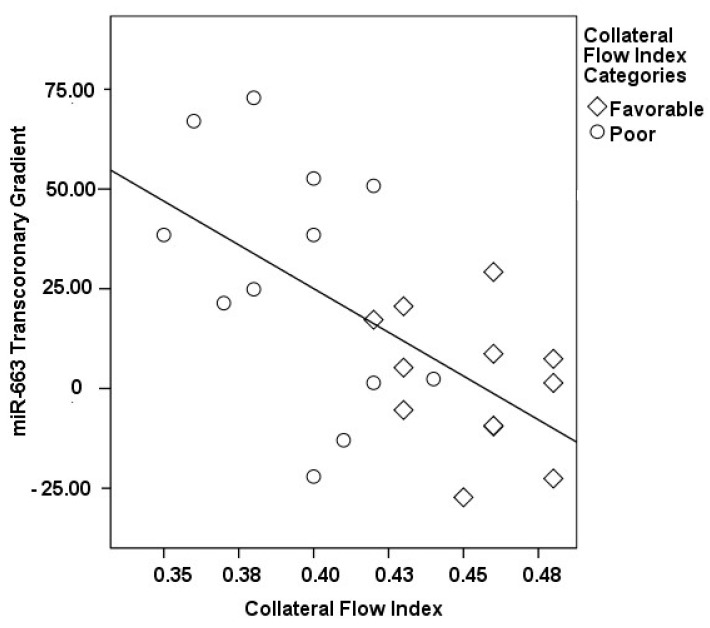
The correlation between the transcoronary gradient of miR-663 and the CFI.

**Figure 9 medicina-60-00590-f009:**
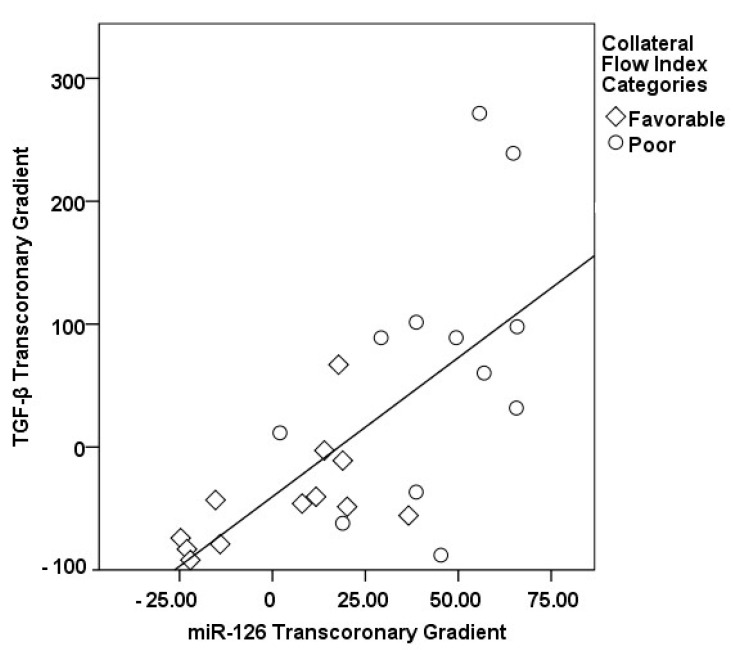
The correlation between the transcoronary gradients of miR-126 and TGF-β.

**Figure 10 medicina-60-00590-f010:**
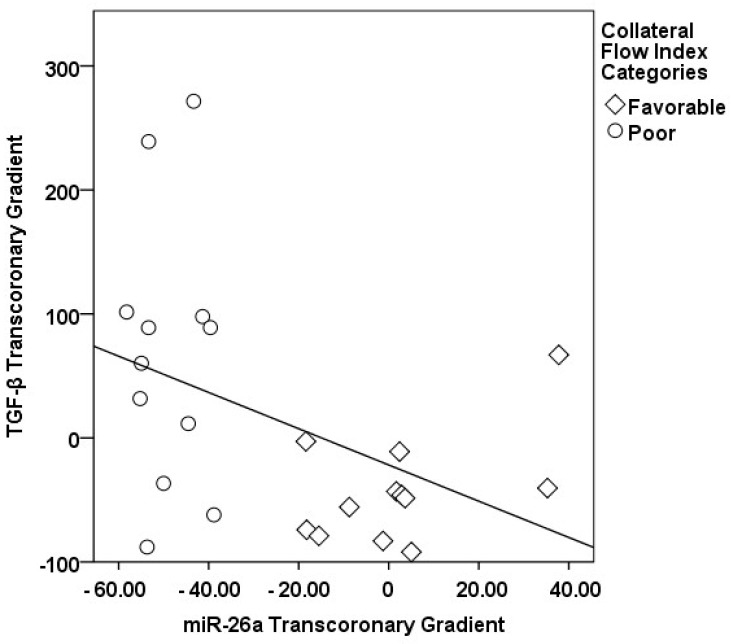
The correlation between the transcoronary gradients of miR-26a and TGF-β.

**Figure 11 medicina-60-00590-f011:**
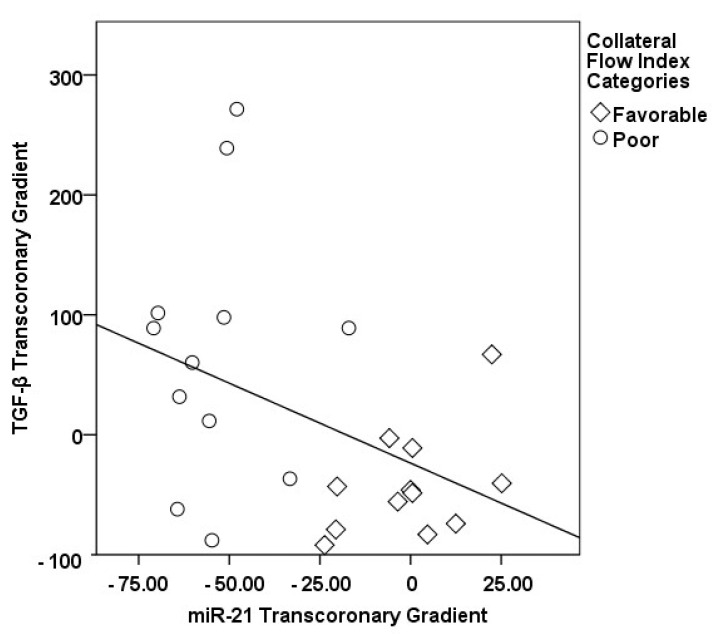
The correlation between the transcoronary gradients of miR-21 and TGF-β.

**Figure 12 medicina-60-00590-f012:**
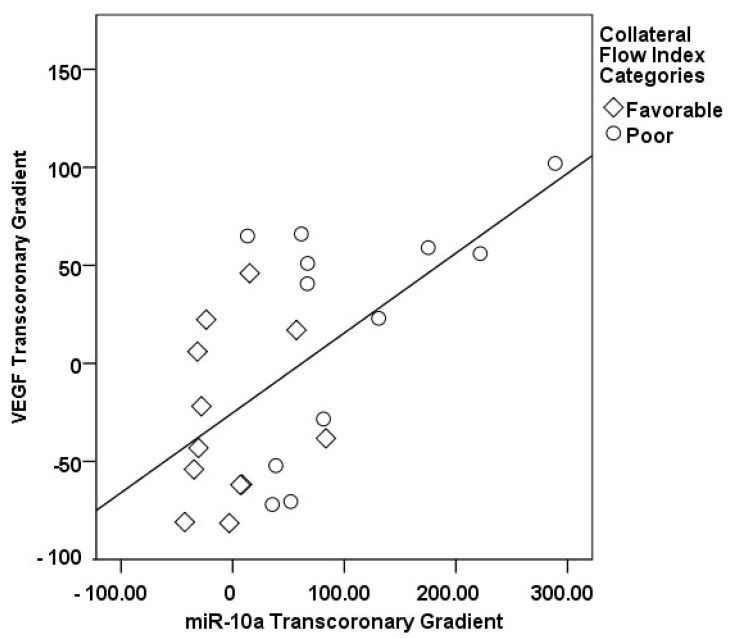
The correlation between the transcoronary gradients of miR-10a and VEGF.

**Figure 13 medicina-60-00590-f013:**
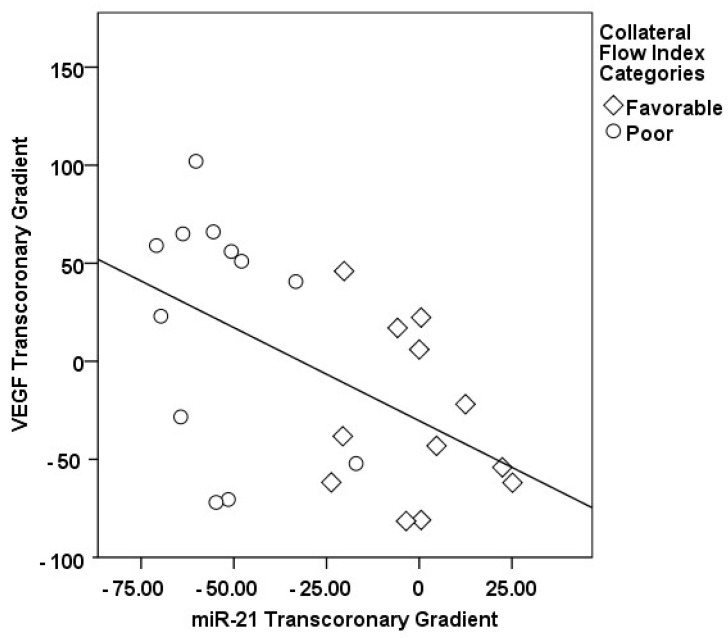
The correlation between the transcoronary gradients of miR-21 and VEGF.

**Figure 14 medicina-60-00590-f014:**
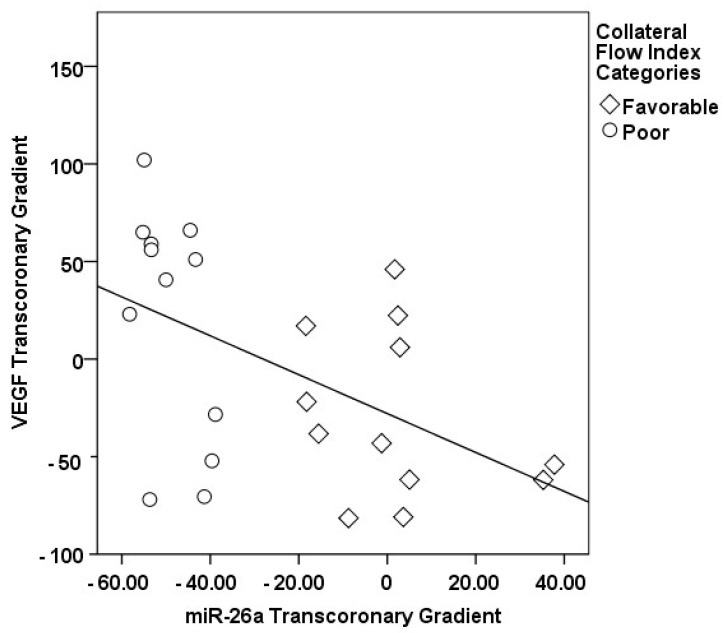
The correlation between the transcoronary gradients of miR-26 and VEGF.

**Table 1 medicina-60-00590-t001:** Baseline demographic, clinical, and biochemical characteristics of the study cohort.

Parameters	Healthy Subjects Group I (n = 30)	CAD without CTO Group II (n = 33)	CTO Patients Group III (n = 63)	*p*-Value
Age, years	54.8 ± 12.0	52.0 ± 11.0	55.3 ± 10.8	0.374
Male, %	80	69.7	81	0.427
BMI, kg/m^2^	28.6 ± 4.0	28.7 ± 4.9	27.9 ± 3.8	0.593
Hypertension, %	60	65	69.4	0.672
Dyslipidemia, %	46.7	63.6	66.7	0.170
Diabetes, %	20	21.2	17.5	0.895
Current smoking, %	53.3	48.5	39.7	0.294
Acetylsalicylic acid, %	10	100	89.9	<0.001
Clopidogrel, %	3.3	21.2	23.8	0.050
Beta blocking agents, %	33.3	57.6	68.3	0.006
Calcium channel blockers, %	13.3	15.2	14.3	0.979
RAAS inhibitors, %	70	66.7	69.8	0.942
Statins, %	46.7	45.5	55.6	0.562
LV EF, %	59.2 ± 6.3	61.5 ± 5.7	59.6 ± 5.9	0.228
LV diastolic dysfunction, %	46.7	48.5	73	0.014
Culprit lesion				
LAD, %	-	33.3	42.4	0.204
Cx, %	-	35.5	31.7	
RCA, %	-	31.2	25.9	
Symptom duration				
3–12 months, %	0	18.2	47.1	<0.001
>12 months, %	0	18.2	45.1	
Ischemic burden				
<5%	100	48.1	21.4	<0.001
5–10%	0	29.6	42.9	
>10%	0	22.2	35.7	
Hemoglobin, g/dL	13.3 ± 1.8	13.0 ± 2.2	13.5 ± 1.8	0.429
Creatinine, g/dL	0.8 ± 0.1	0.8 ± 0.2	0.9 ± 0.3	0.344
Leukocyte count, ×10^9^/L	10.0 ± 2.2	11.5 ± 2.8	10.4 ± 3.5	0.156
NLR	4.4 ± 3.8	4.4 ± 3.0	4.2 ± 5.6	0.983
Glucose, mg/dL	112.5 ± 39.2	121.9 ± 36.5	128.8 ± 47.9	0.285
Total cholesterol, mg/dL	183.5 ± 47.2	198.2 ± 40.2	194.0 ± 42.1	0.377
LDL cholesterol, mg/dL	116.3 ± 28.3	113.0 ± 37.6	109.5 ± 35.8	0.702
HDL cholesterol, mg/dL	36.6 ± 9.2	42.9 ± 13.0	42.1 ± 12.8	0.078
Non-HDL cholesterol, mg/dL	146.9 ± 47.2	155.3 ± 40.8	152.0 ± 41.1	0.733
Remnant cholesterol, mg/dL	29.7 ± 39.0	26.2 ± 34.8	27.1 ± 24.8	0.900
TGF-β, ng/mL	769.8 ± 506.7	1016.0 ± 606.9	967.8 ± 480.7	0.201
IL-1β, ng/mL	4.5 ± 4.3	6.7 ± 4.7	7.6 ± 5.0 ^1^	0.042
MMP-2, ng/mL	464.7 ± 233.2	349.6 ± 194.6	311.9 ± 200.2 ^1^	0.016
IL-10, ng/mL	12.2 ± 7.1	10.4 ± 4.3	12.5 ± 5.8	0.370
VEGF, ng/mL	683.3 ± 501.6	386.5 ± 214.7 ^1^	363.2 ± 211.8 ^1^	0.001
hs-Troponin, ng/mL	0.4 ± 0.9	5.5 ± 14.3	6.3 ± 10.1 ^1^	0.032
NT-pro BNP, pg/mL	166.8 ± 86.6	262.1 ± 154.7	217.0 ± 178.4	0.092
hs-CRP, ng/mL	3.6 ± 4.0	3.1 ± 2.5	3.4 ± 2.1	0.785

Abbreviations: CAD: Coronary Artery Disease; CTO: Chronic Total Occlusion; BMI: Body Mass Index; LV EF: Left Ventricular Ejection Fraction; LAD: Left Anterior Descending Artery; Cx: Circumflex Artery; RCA: Right Coronary Artery; RAAS: Renin–Angiotensin–Aldosterone System; NLR: Neutrophil-to-Lymphocyte Ratio; LDL: Low-Density Lipoprotein; HDL: High-Density Lipoprotein; TGF-β: Transforming Growth Factor Beta; IL: Interleukin; MMP-2: Matrix Metalloproteinase-2; VEGF: Vascular Endothelial Growth Factor; hs-CRP: High-Sensitive C-Reactive Protein; and NT-pro BNP: N-Terminal pro B-Type Natriuretic Peptide. A *p*-value < 0.05 indicates statistical significance; ‘^1^’ indicates *p* < 0.05 against Group I.

**Table 2 medicina-60-00590-t002:** Comparison of demographic, clinical, and laboratory characteristics between favorable and poor collateral groups in CTO patients (Group III).

Parameters	Favorable Collateral (n = 17)	Poor Collateral (n = 19)	*p*-Value
Age, years	55.5 ± 8.2	55.1 ± 12.6	0.907
Male, %	82.4	73.7	0.532
Hypertension, %	70.6	73.7	0.836
Diabetes, %	29.4	10.5	0.153
Dyslipidemia, %	64.7	63.2	0.923
Current smoking, %	41.2	26.3	0.445
RAAS inhibitors, %	64.7	73.7	0.559
Statins, %	52.9	63.2	0.535
Beta blocking agents, %	70.6	68.4	0.888
Calcium channel blockers, %	5.9	26.3	0.101
Acetylsalicylic acid, %	94.1	94.7	0.935
Clopidogrel, %	23.5	15.8	0.558
Culprit Lesion			
LAD, %	42.4	43.2	0.359
Cx, %	11.8	15.8	
RCA, %	45.9	41.1	
Symptom duration			
3–12 months, %	52.9	58.8	0.345
>12 months, %	35.3	41.2	
Ischemic burden			
<5%	21.4	20.0	0.979
5–10%	35.7	33.3	
>10%	42.9	46.7	
CFI	0.45 ± 0.02	0.38 ± 0.03	<0.001
LV EF, %	59.18 ± 6.28	59.37 ± 6.12	0.927
LV diastolic dysfunction, %	70.6	68.4	0.888
Hemoglobin, g/dL	13.57 ± 2.43	13.27 ± 1.71	0.672
Leukocyte count, ×10^9^/L	10.67 ± 4.11	9.68 ± 3.16	0.423
Platelet count, ×10^9^/L	235.53 ± 53.66	231.21 ± 53.54	0.811
Neutrophil, ×10^9^/L	5.04 ± 2.39	4.92 ± 1.40	0.847
Lymphocyte count, ×10^9^/L	1.56 ± 0.56	1.80 ± 0.90	0.385
NLR	5.69 ± 9.89	3.49 ± 1.31	0.356
Glucose, mg/dL	105.18 ± 29.12	137.11 ± 51.59	0.032
Creatinine, mg/dL	0.99 ± 0.44	0.90 ± 0.23	0.447
Total Cholesterol, mg/dL	189.29 ± 38.08	191.00 ± 36.31	0.891
HDL Cholesterol, mg/dL	46.35 ± 15.80	39.68 ± 10.14	0.137
LDL Cholesterol, mg/dL	111.88 ± 26.86	101.47 ± 35.09	0.329
Non-HDL Cholesterol, mg/dL	142.94 ± 35.64	151.32 ± 38.26	0.503
Remnant Cholesterol, mg/dL	21.65 ± 21.43	36.32 ± 30.44	0.108
High-sensitive Troponin, ng/mL	6.63 ± 12.04	5.99 ± 8.40	0.853
NT-Pro BNP, pg/mL	195.75 ± 119.01	269.73 ± 237.12	0.268
hs-CRP, ng/L	3.59 ± 2.24	2.70 ± 2.25	0.257
∆ TGF-β, %	−35.30 ± 49.21	107.33 ± 148.21	0.003
∆ IL-1β, %	−87.09 ± 127.50	−44.73 ± 142.03	0.417
∆ IL-10, %	12.90 ± 35.68	13.11 ± 41.23	0.988
∆ VEGF, %	−27.78 ± 41.39	25.05 ± 52.03	0.006
∆ MMP-2, %	12.78 ± 68.89	−26.81 ± 82.38	0.173
∆ miR-10a, %	−1.98 ± 39.17	102.76 ± 84.65	0.001
∆ miR-19a, %	11.49 ± 55.21	−24.46 ± 9.79	0.037
∆ miR-21, %	−53.30 ± 15.43	−0.69 ± 15.82	<0.001
∆ miR-23b, %	−21.36 ± 23.64	3.49 ± 28.38	0.029
∆ miR-26a, %	−48.88 ± 6.89	2.20 ± 18.23	<0.001
∆ miR-92a, %	54.72 ± 67.35	5.42 ± 33.58	0.033
∆ miR-126, %	44.26 ± 19.96	2.32 ± 20.91	<0.001
∆ miR-130a, %	63.79 ± 37.58	2.98 ± 24.70	<0.001
∆ miR-663, %	27.94 ± 30.87	1.32 ± 16.99	0.016
∆ let7d, %	251.46 ± 84.97	1.10 ± 16.46	<0.001

Abbreviations: RAAS Inhibitors: Renin–Angiotensin–Aldosterone System; LAD: Left Anterior Descending Artery; Cx: Circumflex Artery; RCA: Right Coronary Artery; CFI: Collateral Flow Index; LVEF: Left Ventricular Ejection Fraction; NLR: Neutrophil-to-Lymphocyte Ratio; HDL: High-Density Lipoprotein; LDL: Low-Density Lipoprotein; NT-pro BNP: N-Terminal pro-Brain Natriuretic Peptide; hs-CRP: High-Sensitive C-Reactive Protein; TGF-β: Transforming Growth Factor Beta; IL: Interleukin; VEGF: Vascular Endothelial Growth Factor; and MMP-2: Matrix Metalloproteinase-2. The ∆ symbol is employed to signify the transcoronary gradient, quantifying the difference in the concentration of a given biomarker between the aorta and the coronary sinus, which may reflect the substance’s uptake or release within the myocardium. A *p*-value < 0.05 indicates statistical significance.

## Data Availability

No new data were created or analyzed in this study. Data sharing is not applicable to this article.

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
