# Peer review of "Transcoronary Gradients of Mechanosensitive MicroRNAs as Predictors of Collateral Development in Chronic Total Occlusion"

_medicina, 2024, doi:10.3390/medicina60040590_

Round 1
Reviewer 1 Report
Comments and Suggestions for Authors
The manuscript is clear, relevant to the field, and presented in a well-structured manner. Most references cited are extremely pertinent and within the last five years.
The manuscript is scientifically sound, and the experimental design is appropriate to test the hypothesis. This two-center observational study examined the association between mechano-miRs and collateral vessel development in CTO patients. The objective was to understand the molecular mechanisms driving collateral circulation in CTO. Of 810 patients undergoing coronary angiography, 126 were selected based on stringent criteria. These included excluding those with a history of myocardial infarction, left ventricular ejection fraction below 50%, or previous coronary artery bypass grafting.
The supplied tables are acceptable. The facts were presented well and were simple to analyze and comprehend. However, figures are not included, which would make the text more comprehensible.
A sample for convenience is considered to have been taken because the computation for the sample was not carried out. More clarity was needed on the selection process for healthy individuals and whether they were subjected to all testing. Even though there was an extensive debate that might have been more objective and clearer, the conclusions drawn were weakened because of the research's limitations, which showed the challenges in the findings.
Author Response
Response to Reviewer 1 Comments
Dear Reviewer,
We express our deepest appreciation for the comprehensive review you have provided and the invaluable insights shared regarding our manuscript. Your feedback has been instrumental in enhancing the quality and clarity of our research. Please find below our responses to your comments, which have guided our revisions and updates to the manuscript.
Comment 1: A sample for convenience is considered to have been taken because the computation for the sample was not carried out.
Response 1: In the conducted research, the determination of the requisite sample size was meticulously executed using robust statistical methodologies to ensure the adequacy of the study's power. This was crucial for the ability to discern statistically significant differences in the levels of mechano-miRs between cohorts of CTO patients characterized by either favorably or poorly developed collateral circulation. Based on effect sizes and alpha levels customary in clinical research, the power analysis indicated the necessity of enrolling a minimum of 18 participants per group. This requirement was to achieve the statistical power sufficient to unequivocally determine the influence of mechano-miRs on collateral vessel development within the context of CTO. To meet this objective and capture the effect within a comprehensive sample of 36 CTO patients, a strategy was formulated to examine a total of 120 patients suffering from CTO. The implementation of this plan was carefully aligned with the operational capacities and throughput of our institutions’ catheterization laboratories. Consequently, the review of 810 consecutive coronary angiography procedures was initiated. This methodological framework was supported by a detailed power analysis, utilizing conventional statistical benchmarks such as a standard alpha level of 0.05 and a power setting of 80%. These specifications play a critical role in minimizing the risk of Type I and Type II errors, thus strengthening the validity of the research findings.
In concordance with your feedback, we have updated the manuscript to reflect a more precise articulation of our methodological approach. The revised text, now found on page 3, paragraph 6, lines 148-154, reads: ‘A power analysis, adhering to clinical research norms with effect sizes and an alpha level of 0.05, underscored the need for at least 18 participants in each group to robustly ascertain the impact of mechano-miRs on collateral development in CTO patients. Consequently, a meticulously designed approach led to the evaluation of 120 CTO patients, aligning with our catheterization labs' capabilities, and facilitated the review of 810 consecutive angiographies, achieving an 80% power threshold for conclusive insights.’
Comment 2: More clarity was needed on the selection process for healthy individuals and whether they were subjected to all testing.
Response 2: In the present study, adherence to ethical guidelines necessitated the exclusion of transcoronary gradient measurements within our cohort of healthy volunteers. Instead, as a viable alternative, we opted to analyze serum biomarkers through the collection of venous blood samples from this group. In contrast, our methodology for patients diagnosed with CTO encompassed a comprehensive suite of evaluations, including both transcoronary gradients and the collateral flow index, executed with meticulous attention to detail.
Under your invaluable critique, we have taken the initiative to make a pertinent revision to our text. Specifically, we have appended the following statement to our manuscript on page 3, paragraph 3, lines 119-121: 'In our investigation, ethical considerations precluded the execution of transcoronary gradient measurements within the cohort of healthy volunteers.'
Comment 3: Even though there was an extensive debate that might have been more objective and clearer, the conclusions drawn were weakened because of the research's limitations, which showed the challenges in the findings.
Response 3: Regarding your concerns about the limitations of our research and how they might have affected the strength of our conclusions, we acknowledge the challenges you highlighted. The nature of scientific inquiry often involves navigating through complexities and constraints, which can indeed impact the robustness of findings. However, we have implemented several measures to address these limitations and ensure the reliability and validity of our study's outcomes.
We have employed rigorous statistical analyses to mitigate the impact of the study's inherent limitations. Through detailed power analysis and careful selection of statistical methods, we aimed to maximize the precision of our findings despite the constraints faced. We have strived for utmost transparency in detailing our methodology, data analysis, and interpretation of results. By comprehensively reporting the steps taken and the rationale behind them, we believe the research community can critically assess the merit of our work and its implications. In the discussion section, we have contextualized our findings within the broader landscape of existing literature. This not only situates our study within ongoing scientific discourse but also highlights its novel contributions despite the acknowledged limitations. We have outlined potential avenues for future research that can build upon our findings and address the limitations noted. This includes suggestions for studies with expanded sample sizes, diverse populations, and the application of alternative methodological approaches to validate further and extend our conclusions.
In conclusion, while we recognize the constraints of our study, we have endeavored to conduct our research with the highest level of academic integrity and methodological rigor. Our findings, despite the limitations, contribute valuable insights to the field and lay a foundation for future investigations to explore.
We hope that our revisions and responses adequately address your concerns. We are grateful for the opportunity to improve our manuscript under your guidance and look forward to any further feedback you may have.
Sincerely,
Mustafa Gökhan Vural, M.D.
University of Health Science

Reviewer 2 Report
Comments and Suggestions for Authors
Congratulations to the authors on preparing the manuscript which presents a study investigating the influence of mechanosensitive microRNAs (mechano-miRs) on collateral development in chronic total occlusion patients.
The study is well presented, minor comments
1. Was ethics approval requested for this study? Please include the ethics approval details ie institutional/committee name and approval number
2. For better clarity I suggest including flowchart of included and excluded participants
Author Response
Response to Reviewer 2 Comments
Dear Reviewer,
We express our deepest appreciation for the comprehensive review you have provided and the invaluable insights shared regarding our manuscript. Your feedback has been instrumental in enhancing the quality and clarity of our research. Please find below our responses to your comments, which have guided our revisions and updates to the manuscript.
Comment 1: Was ethics approval requested for this study? Please include the ethics approval details ie institutional/committee name and approval number
Response 1: The ethical clearance for this investigation was rigorously sought and duly received, signifying the study's adherence to the highest standards of research ethics. The study received generous support from The Scientific and Technological Research Council of Turkey (TÜBÄ°TAK). Ethical approval was conferred by the Ethical Committee of Sakarya University, with the formal endorsement dated February 24, 2016, under the approval identification number 16214662/050.01.04/53. This authorization underscores our commitment to conducting research within the ethical frameworks and guidelines established by the scientific community.
Comment 2: For better clarity I suggest including flowchart of included and excluded participants
Response 2: We have taken your suggestion into account and decided to enhance the clarity of our manuscript by incorporating a flowchart that delineates the participant inclusion and exclusion criteria. To this end, we have designated this flowchart as Figure 1 and positioned it at the end of the manuscript for easy reference (page 2, paragraph 5, line 99).
We hope that our revisions and responses adequately address your concerns. We are grateful for the opportunity to improve our manuscript under your guidance and look forward to any further feedback you may have.
Sincerely,
Mustafa Gökhan Vural, M.D.
University of Health Science

Reviewer 3 Report
Comments and Suggestions for Authors
Enjoyable and elegant study. Following are my few suggestions/observations.
Lines 23-24: explain the meaning of the acronyms (TGF-β and VEGF) the first time you use them. Or just write them in extensive, since these terms do not recur anymore in the Abstract.
Lines 199-200: add “…in CTO patients (Group III)” at the end of Table 2’s legend.
Lines 229 and 247: I can’t find any figure along your manuscript.
Author Response
Response to Reviewer 3 Comments
Dear Reviewer,
First and foremost, we extend our deepest gratitude for the time and effort you have dedicated to reviewing our manuscript. Your insights and suggestions are invaluable to enhancing the quality and clarity of our work. We are pleased to address the points you have raised, as follows:
Comment 1: Lines 23-24: explain the meaning of the acronyms (TGF-β and VEGF) the first time you use them. Or just write them in extensive, since these terms do not recur anymore in the Abstract.
Response 1: We appreciate your suggestion to elucidate the acronyms TGF-β (Transforming Growth Factor Beta) and VEGF (Vascular Endothelial Growth Factor) upon their first mention in the abstract. Recognizing the importance of clarity for our readers, we have revised the text to include the full terms, followed by the acronyms in parentheses.
Comment 2: Lines 199-200: add “…in CTO patients (Group III)” at the end of Table 2’s legend.
Response 2: Your observation regarding the necessity to specify the patient group in Table 2's legend is well-taken. We have updated the legend to include "...in CTO patients (Group III)," thereby providing clearer context and enhancing the table's interpretative value.
Comment 3: Lines 229 and 247: I can’t find any figure along your manuscript.
Upon further examination, it has become evident that an oversight occurred not during the initial submission, but rather in the integration of figures into the manuscript's main body. The figures were indeed submitted separately through the submission system, but due to a miscommunication, they were not correctly appended within the manuscript document itself. We have now thoroughly reviewed the submission system and confirmed that all figures were correctly uploaded and are available. To rectify this oversight, we have taken immediate action to ensure that all figures are now appropriately incorporated at the end of the manuscript document. This adjustment allows for a complete and accurate presentation of our work, including the visual aids crucial for understanding our research findings.
We hope that our revisions and responses adequately address your concerns. We are grateful for the opportunity to improve our manuscript under your guidance and look forward to any further feedback you may have.
Sincerely,
Mustafa Gökhan Vural, M.D.
University of Health Science

Reviewer 4 Report
Comments and Suggestions for Authors
The figures (1-13) are missing in the submitted manuscript. Could you please provide the figures with their respective legends?
Author Response
Response to Reviewer 4 Comments
Dear Reviewer,
First and foremost, we extend our deepest gratitude for the time and effort you have dedicated to reviewing our manuscript. Your insights and suggestions are invaluable to enhancing the quality and clarity of our work. We are pleased to address the points you have raised, as follows:
Comment 1: The figures (1-13) are missing in the submitted manuscript. Could you please provide the figures with their respective legends?
Response 1: Upon further examination, it has become evident that an oversight occurred not during the initial submission, but rather in the integration of figures into the manuscript's main body. The figures were indeed submitted separately through the submission system, but due to a miscommunication, they were not correctly appended within the manuscript document itself. We have now thoroughly reviewed the submission system and confirmed that all figures were correctly uploaded and are available. To rectify this oversight, we have taken immediate action to ensure that all figures are now appropriately incorporated at the end of the manuscript document. This adjustment allows for a complete and accurate presentation of our work, including the visual aids crucial for understanding our research findings.
We hope that our revisions and responses adequately address your concerns. We are grateful for the opportunity to improve our manuscript under your guidance and look forward to any further feedback you may have.
Sincerely,
Mustafa Gökhan Vural, M.D.
University of Health Science
